## Perspective

DNA; transcritpion; fluorescence; RNA; protein–nucleic acid interactions; translation; synthetic biology

**Corresponding author:**
Gijs J. L. Wuite;
Email: g.j.l.wuite@vu.nl

# Challenges in observing transcription–translation for bottom-up synthetic biology

Vadim G. Bogatyr [ID] and Gijs J. L. Wuite [ID]

Department of Physics and Astronomy and LaserLab, Faculty of Science, Vrije Universiteit, Amsterdam, Netherlands

## Abstract

Synthetic biology aims to create a viable synthetic cell. However, to achieve this goal, it is essential first to gain a profound understanding of the cellular systems used to build that cell, how to reconstitute those systems in the compartments, and how to track their function. Transcription and translation are two vital cellular systems responsible for the production of RNA and, consequently, proteins, without which the cell would not be able to maintain itself or fulfill its functions. This review discusses in detail how the Protein synthesis Using Recombinant Element (PURE) system and cell lysate are used to reconstitute transcription–translation in vitro. Furthermore, it examines how these systems can be encapsulated in GUVs using the existing methods. It also assesses approaches available to image transcription and translation with a diverse arsenal of fluorescence microscopy techniques and a broad collection of probes developed in recent decades. Finally, it highlights solutions for the challenge ahead, namely the decoupling of the two systems in PURE, and discusses the prospects of synthetic biology in the modern world.

## Introduction

Over the past decades, our knowledge of living cells' inner molecular and biochemical workings has reached a formerly unimaginable depth. Nevertheless, despite all the novel techniques, scientific discoveries, and advances, the living cells remain a mystery, particularly its *living* aspect (Schrödinger 1944). Thus, one of the significant scientific challenges of this century is to uncover what creates a living artificial cell (Ehmoser-Sinner and Tan 2018). The current strategies toward this goal lie on the spectrum between two drastically different approaches: top-down and bottom-up.

In the top-down approach, a living cell is stripped of all unnecessary, that is, non-vital parts. A prime example of this approach was the mutagenesis study of the *Mycoplasma mycoides* genome, where only 473 of the 901 genes were kept to produce a minimal synthetic JCVI syn3.0 cell (Hutchison *et al.* 2016). However, the fact that 79 out of those 473 genes remain unassigned highlights the fundamental property of living systems: *the whole is more than the sum of its parts.* The bottom-up approach in synthetic biology tackles this by taking an opposite rationalistic and somewhat mechanical approach to reconstituting cellular functions, properties, and structures. Building blocks, natural or synthetic cellular components, are assembled and combined to create a system that could be considered *living*. As scientists focus on particular cellular systems and develop new biophysical techniques to interact with them, they also gain a profound understanding of how these systems work and converge with other building blocks.

One of the essential systems required to achieve a synthetic cell is the processing of the genetic material, DNA, and protein expression. The latter is necessary to guarantee the function of maintaining itself by recreating its own parts, which is typical for living systems (Maturana and Varela 1972). Without the transcription and translation systems, the synthetic cell cannot produce necessary metabolites, divide, and grow, that is, fulfill all fundamental criteria of the minimal viable organism.

Another requirement imposed for a prospective synthetic cell to function like its natural counterparts is the boundary, separating it from the environment. The selective permeability of the cellular membrane allows for extensive regulation and control of the interior processes. For this reason, attempts to produce synthetic cells and study cellular systems *in vitro* commonly involve encapsulation (Noireaux and Libchaber 2004) in giant unilamellar vesicles (GUVs) (Van de Cauter *et al.* 2023). GUVs allow the reproduction of spatial separation conditions and can be produced in various ways, which will be discussed in this review, along with the implications of transcription–translation encapsulation, its products, and the means to image those processes, as well as the challenges that arise to do so.

Finally, when it comes to storing, processing, and inheriting genetic information by the proto-synthetic cells, one way is to encapsulate multiple copies of the same chromosome or a set of plasmids (Nordström and Austin 1989). In this case, random portioning will create offspring cells with some copies of the necessary genetic information. However, over the generations, this will lead

to an increasing heterogeneity in the number of gene copies (Huh and Paulsson 2011). Additionally, these chromosomes, containing 100s of genes, would take increasingly more space and energy. An alternative nature-inspired approach is to encapsulate a single chromosome and design the system to yield single-chromosome offspring cells, for example, through active (Lemon and Grossman 2001) or entropy-driven separation of chromosomes (Jun and Mulder 2006) before the division (Olivi *et al.* 2021). Favoring the second approach, we highlight techniques that allow single-molecule transcription tracking at low DNA copy numbers in GUVs.

## Achieving transcription–translation *in vitro*

Two major yet conceptually diametrical approaches for *in vitro* transcription–translation are using cell lysate (Didovyk *et al.* 2017) and the Protein synthesis Using Recombinant Element (PURE) system (Shimizu *et al.* 2001, 2005). Cell lysate can be viewed as the top-down solution, capturing all the necessary components to achieve transcription–translation *in vitro* as they are present in live bacteria, commonly *Escherichia coli* lysate is used for this purpose (Didovyk *et al.* 2017), coming with innately coupled transcription and translation systems, thus ensuring high yields (Hansen *et al.* 2016b). Furthermore, crowding in the solution emulates a native intracellular environment and can improve yields (Vibhute *et al.* 2020). Finally, the use of different lysate dilutions allows researchers to regulate the kinetic rates of mRNA production and degradation. The use of lysate does come at a cost: RNases and proteases are present and active in the lysate solution. They are actively working against the transcription and translation systems, digesting synthesized RNA and protein molecules. It is because of the previously mentioned coupling of the transcription–translation system that this potential issue is mitigated. Once the systems are decoupled, the yields decrease significantly (Hansen *et al.* 2016b). There is virtually no control over the composition of the lysate, making it essentially a black box where some of the cellular pathways remain active while others cease. This lack of information about components was partially circumvented by using mass-spectrometry to reveal the detailed proteome of the lysate (Foshag *et al.* 2018). Additionally, theoretical models for lysate are being developed, such as a system of ordinary differential equations that was used to describe the expression of the reporter gene. Experimental data supplemented this system and showed the regulatory effects of various promoters and ribosome-binding sites (Marshall and Noireaux 2019). Nevertheless, more studies of the cell lysate composition and fine-tuned pathway models are needed to disentangle the mesh of metabolic interaction inside it. This data could then be applied to artificially enhance specific pathways by supplementing additional metabolites depending on the needs (Miguez *et al.* 2021).

PURE system was initially engineered bottom-up through the combination of T7 bacteriophage (Figure 1) RNA polymerase responsible for transcription, the *E. coli* translation machinery, and necessary metabolites to provide building blocks for RNA and protein (Shimizu *et al.* 2001, 2005). All its components, including the choice of salts, and their concentrations are listed in the Supplementary section. The composition of PURE was highly optimized, making it possible to create a kinetic description of the PURE system (Doerr *et al.* 2019). Though it is possible to produce all PURE system components and solutions in the lab, it is far more complex than lysate production since ribosome isolation and over 36 protein purifications are required (Grasemann et al. 2021). Therefore, several commercial PURE system kits are currently available, optimized for

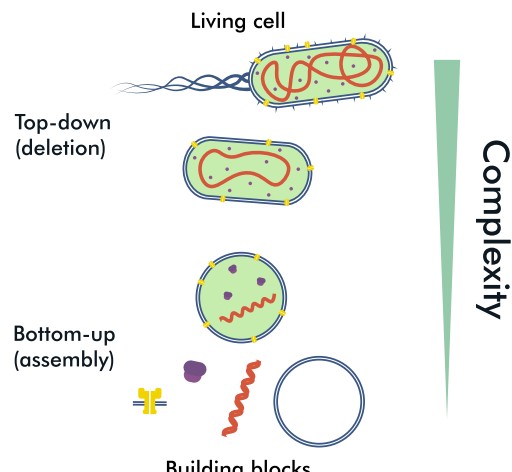

**Figure 1.** Two diametrically opposite approaches for synthetic cell production. The top-down approach aims to decrease the complexity of the system by discarding unnecessary elements, while the bottom-up approach combines building blocks to increase the complexity of the system.

different applications, and sold by GeneFrontier (Japan), New England Biolabs (USA), and Creative Biolabs (USA). The precise control over the content minimizes any RNase and protease activity, which the manufacturers claim to reduce 'greatly'. However, the combination of transcription and translation complexes from two different, though compatible, organisms makes the PURE system underperform by at least an order of magnitude compared with cell lysate when it comes to protein expression (Hillebrecht and Chong 2008). Ultimately, the choice of PURE or lysate comes down to the particular application, protein of interest, and technical constraints. We provide their overview comparison in Table 1.

In the context of synthetic cells, independent of which approach is used for *in vitro* transcription and translation, the end goal is not just to maximize protein yields but to reconstitute cellular systems, gene interactions, and metabolic pathways in a coordinated manner (Noireaux *et al.* 2003; Shin and Noireaux 2012). This is somewhat problematic, considering that fluorescent proteins, most commonly used to study *in vitro* expression, are great for reporting but play no functional role. An alternative to that is the use of transcription–translation systems to produce proteins involved in

**Table 1.** Comparison of cell lysate and PURE system for in vitro transcription–translation in the context of bottom-up synthetic cells

|  | Cell lysate | PURE system |
|---|---|---|
| Number of components | 1000+ | 36 (can vary) |
| Transcription system origin | *E. coli* | T7 |
| Translation system origin | *E. coli* | *E. coli* |
| Transcription–translation coupling | Yes | No |
| Protein yield (relative to each other) | 1x | 0.001–0.01x |
| Protease and nuclease activity | Yes | Minimal to none |
| Cost (commercial product) | €0.4–0.6/μL | €0.5–1.25/μL |

*Note:* Cell lysate costs were calculated using Biotechrabbit (Germany) and Cube Biotech (Germany) catalogs; (Nirenberg and Matthaei 1961; Hansen et al. 2016b; Didovyk et al. 2017; Foshag et al. 2018; Marshall and Noireaux 2019). PURE system costs were calculated using GeneFrontier (Japan) and New England Biolabs (USA) catalogs and their distributors in the Netherlands; (Shimizu et al. 2001, 2005; Doerr et al. 2019; Cauter et al. 2021; Grasemann et al., 2021), reviews (Li et al. 2017; Laohakunakorn et al. 2020; Garenne et al. 2021a; Cui et al. 2022).

**Table 2.** Comparison of GUV production methods for in vitro transcription and translation

| Method | Size distribution | Size control | Encapsulation efficiency | Min. sample volume | Equipment requirements |
|---|---|---|---|---|---|
| cDICE | + | ++ | ++ | >100 µL | € € € |
| eDICE | + | + | ++ | 2 µL | € € |
| Inverted emulsion | + | + | ++ | 2 µL | € |
| Lipid-coated glass beads | – | – | + | 2 µL | € |
| Microfluidic methods | ++ | +++ | +++ | >100 µL | € € € € |

*Note:* **Size distribution:** – polydisperse, + broad monodisperse, ++ narrow monodisperse**; Size control:** – no size control; + limited empirical control; ++ limited mechanical control; +++ precise control; **Encapsulation efficiency:** + low, ++ moderate, +++ high; **Equipment requirements:** cDICE – syringe pump, specially-designed capillary stand, specially-designed rotating stage with chamber holder, 3D printed chamber, glove box for improved yields (Abkarian et al. 2011; Blosser et al. 2016; Bashirzadeh et al. 2021b, 2021a; Cauter et al. 2021, 2024; Litschel et al. 2021); eDICE – specially-designed rotating stage with chamber holder, 3D printed chamber, glove box for improved yields (Baldauf et al. 2023a, 2023b; Wubshet et al. 2023); Inverted emulsion – centrifuge (Nishimura et al. 2012; Soga et al. 2014; Litschel et al. 2018; Berhanu et al. 2019; Moga et al. 2019; Yoshida et al. 2019; Garenne and Noireaux 2020; Zhang et al. 2023); Lipid-coated glass beads – none (Nourian et al. 2012; Tanasescu et al. 2018; Blanken et al. 2020; Kattan et al. 2021; Gonzales et al. 2022); Microfluidic methods – microfluidic pressure-driven flow controller, microscope to observe production, clean room as the manufacturing of narrow-channeled PDMS chips is highly complex (Teh et al. 2011; Yan et al. 2013; Deshpande et al. 2016; Kang et al. 2016; Deng et al. 2018; Vibhute et al. 2020; Yandrapalli et al. 2021; Gonzales et al. 2022).

cellular functions such as lipid synthesis (Scott *et al.* 2016), cytoskeleton formation (Cauter *et al.* 2021; Litschel *et al.* 2021), or genome replication (Van Nies *et al.* 2018). Another possibility is to design systems with genetic circuits, such as in the case of the TX-TL Toolbox (myTXTL is the commercial name), which is based on *E. coli* lysate (Shin and Noireaux 2012; Garamella *et al.* 2016; Garenne *et al.* 2021b). This system was numerically described (Marshall and Noireaux 2019), characterized by mass spectrometry (Garenne *et al.* 2019), and used to create biocircuits made of sigma factor-regulated plasmids (Garamella *et al.* 2016; Agrawal *et al.* 2019). Studying how these synthetic cell-tailored transcription and translation systems are reconstituted inside GUVs is particularly relevant, which is why compatible GUV production methods will be the focus of the next section.

## Encapsulating transcription–translation in GUVs

The methods for GUV production are generally divided into two categories: swelling-based and emulsion-based approaches. The swelling-based approaches involve the rehydration of lipids on, among other things, gel (Weinberger *et al.* 2013), electrode (Angelova and Dimitrov 1986), and glass beads (Nourian *et al.* 2012; Tanasescu *et al.* 2018). These methods offer limited encapsulation efficiency, and only the lipid-coated glass beads method has been repeatedly and successfully used for *in vitro* transcription and translation. Hence, we only consider this approach in our table GUV production methods (Table 2). It does not offer size control nor produce a monodisperse GUV population, but the simplicity and cost-effectiveness of the method make it very easy to reproduce in the lab. The second large category, the emulsion-based methods, includes diverse methods that produce a lipid bilayer at the most fundamental level by fusing two lipid monolayers. The first one surrounds the water-in-oil droplet with the encapsulated solution, and the second one is on the oil–water interface, through which that droplet passes. In practice, this can be achieved by various approaches of increasing complexity: from the centrifugation in the inverted emulsion method (Nishimura *et al.* 2012; Moga *et al.* 2019) and the use of 3D printed spinning chamber with a water–oil interface, into which the encapsulated solution is pumped through a capillary in cDICE (Abkarian *et al.* 2011; Cauter *et al.* 2021) to the clean-room manufacturing of intricate PDMS chips in microfluidic methods, such as OLA (Deshpande *et al.* 2016), double-emulsion dewetting (Yan *et al.* 2013; Kang *et al.* 2016; Deng *et al.* 2018) and

surfactant-free microfluidic approach (Yandrapalli *et al.* 2021). These microfluidic methods could be viewed as automating a more manual inverted emulsion and making use of surface tension differences to shed the oil phase from the water-in-oil droplet and produce GUVs.

Continuing with the analogy, cDICE, with its microfluidic capillary and a macrochamber, lies somewhere in between. Another similar method, eDICE, combines manual mechanical agitation to produce water-in-oil droplets of the inverted emulsion and the spinning chamber with a water–oil interface of cDICE. Notably, at this time, no peer-reviewed papers have demonstrated *in vitro* transcription and translation using eDICE. However, the successful use of both inverted emulsion and cDICE, moderate encapsulation efficiency, and low costs of necessary components make eDICE a promising approach for future research. Additionally, emulsion-based methods that employ centrifugation of a tube with a specially designed microfluidic insert, such as droplet-shooting and size-filtration (DSSF) (Morita *et al.* 2015) and modification (Venero *et al.* 2022; Deich *et al.* 2023), have not been included due to the limited publications on them.

While all emulsion-based approaches generate monodisperse GUV populations (Figure 2), only the microfluidic approaches ensure highly monodisperse and continuous GUV production. The same could be said about the size control, except that in cDICE, different diameter capillaries offer limited GUV size control. In contrast, the sizes resulting from inverted emulsion/eDICE could only be adjusted to a limited degree empirically by changing the number of times the tube is dragged along a rack to agitate the solutions and mix the encapsulated water-based solution into the oil. The encapsulated volume for the inverted emulsion, eDICE, and lipid-coated glass beads could be as small as a few micrometers,

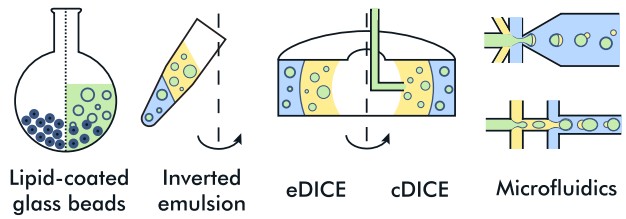

**Figure 2.** Examples of GUV production methods used to encapsulate transcription-translation reaction. The solutions used in each of them are color-coded: encapsulated solution (green), lipids diluted in oil phase (yellow), and exterior aqueous phase (blue).

while cDICE and microfluidic methods require >100 μL samples to be used each time because of the inner volume of the tubing. This volume loss could be problematic when using costly solutions such as commercial PURE systems (1€/μL). Costs are the downside of the microfluidic methods, which generally require more expertise as well as special facilities and equipment to establish them in the lab. While they allow precision manufacturing of GUVs, other less complex methods we listed could be adopted in a shorter time by more different kinds of labs. Furthermore, the comparison table contains several non-numerical but comparative columns, such as Encapsulation efficiency and size control. The challenges of GUV synthesis are not only in choosing the best method and protocol for one's needs but also in ensuring it works optimally in the conditions of a specific lab, as even environmental factors such as air humidity can affect the resulting yields (Cauter *et al.* 2021). For an extensive overview of GUV requirements and production methods for synthetic cell applications beyond *in vitro* transcription and translation, consider the review (Van de Cauter *et al.* 2023).

## Imaging transcription *in vitro*

After discussing the methods to encapsulate the *in vitro* transcription–translation mixture in GUVs, we next examine the ways to follow those two processes in bulk and inside GUVs. In particular, we highlight approaches with single-molecule resolution, which are or could be of use in synthetic cell studies. Among the detection approaches, reverse-transcriptase quantitative PCR (RT qPCR) variations are the go-to method for mRNA studies (Park and Magan 2011; Green and Sambrook 2018; Sato *et al.* 2022). Meanwhile, mass spectrometry offers unparalleled precision when it comes to protein quantification and sensitivity sufficient even to resolve posttranslational modifications (Gerber *et al.* 2003). Both, however, are the end-of-reaction methods used to calculate the quantity of mRNA or protein after the process is finished unless aliquoting is used. For real-time approaches, it is common to employ fluorescent probes that activate upon binding to the mRNA.

One such probe is called a molecular beacon (Tyagi and Kramer 1996; Giesendorf *et al.* 1998; Tsourkas *et al.* 2003a; Goel *et al.* 2005; Bratu *et al.* 2011). It is typically a 20+ oligonucleotide loop, functionalized with a fluorophore and a quencher at opposing ends. Molecular beacons are designed to stay closed in the solution because of the interaction of typically 4–6 oligos-long double-stranded stem segments and open up when its target-binding loop oligos complementary sequence on the mRNA. The third state of the molecular beacon is the open-in-solution state, which produces an intense background signal of unquenched dye and is the major downside of this probe. Another downside is that molecular beacons, being single-stranded ribonucleotides, are prone to degradation by nucleases unless particular chemical strategies, such as the use of 2′-O-methyl ribonucleotides, are used (Majlessi *et al.* 1998). The exact structure, stem length, and fluorophore of the MB have to be optimized for a particular application. For example, having a longer G-C-rich stem improves the selectivity of the probe and results in a lower background signal at the cost of hybridization rate and vice versa (Tsourkas *et al.* 2003a) unless some of the neck segment is also involved in target-binding (Tsourkas *et al.* 2002).

Molecular beacons offer the benefits of bright and stable commercial dyes, which allow the development of various enhanced imaging strategies, particularly for *in vivo* applications (Mao *et al.* 2020), but potentially useful *in vitro* as well. These include using donor and acceptor molecular beacons with a FRET dye pair for the minimization of background noise (Bratu *et al.* 2003; Tsourkas *et al.*

2003b). Other approaches focus on using multiple molecular beacon repeats to achieve better signal-to-noise ratio and even single-molecule resolution for the localization of mRNA molecules. In one study, the localization precision was comparable to that of smFISH, starting with 8x molecular beacon repeats, which came at the cost of significant elongation of the construct. Notably, the molecular beacon-tagged RNA molecules, in this case, were sufficiently fluorescent for imaging by conventional widefield fluorescence microscopy (Chen *et al.* 2017). Meanwhile, another *in vitro* study utilized confocal microscopy and 32x molecular beacon repeats to showcase how crowding in picoliter droplets leads to mRNA concentration in individually resolved spots instead of homogeneous distribution observed in the absence of a crowding agent (Hansen *et al.* 2016a).

Aptamers (Ellington and Szostak 1990) are oligonucleotides or polypeptides developed to bind a specific target ligand. One particular type of aptamers that is of interest for the scope of this review is called fluorescent light-up aptamers (Bouhedda *et al.* 2017). These on-when-bound oligonucleotide aptamers are developed using the Systematic Evolution of Ligands by EXponential enrichment (SELEX) process (Tuerk and Gold 1990), which enables scanning through large libraries of sequences for the one with the highest affinity and specificity for the target ligand. Among them, Spinach, a 98-nucleotide RNA aptamer, was developed as a mimic of Green Fluorescent Protein (GFP), where the interaction of oligos forms a G-quadraplex with affinity for synthetic dye 3,5-difluoro-4-hydroxybenzylidene imidazolinone (DFHBI). This interaction creates a very similar atomical motive to that of amino acids in GFP (Pothoulakis *et al.* 2014), enabling widefield fluorescence imaging of the produced RNA molecules in solution. The stabilization of the fluorophore by the tertiary aptamer structure increases its fluorescence by preventing premature decay from the excited state. To improve the thermal properties and stability of Spinach aptamer, various modifications (Strack *et al.* 2013; Warner *et al.* 2014; Zhang *et al.* 2015; Autour *et al.* 2016) were developed along with Broccoli (Filonov *et al.* 2014; Okuda *et al.* 2017; Kartje *et al.* 2021).

Meanwhile, other wavelengths and fluorophores were used to produce aptamers (Figure 3), such as a single-molecule oriented

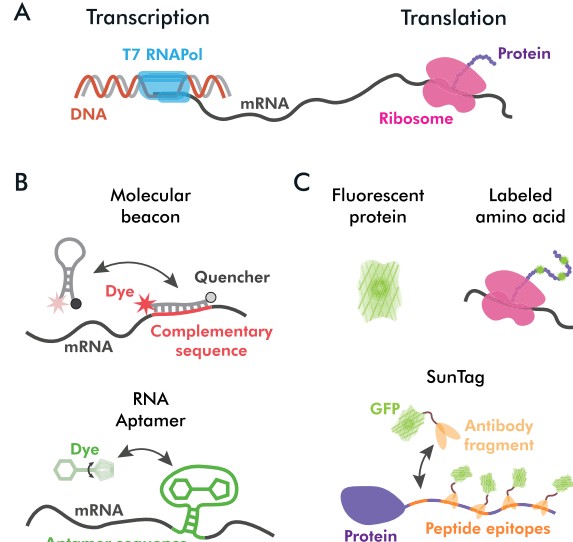

**Figure 3.** (A) Schematic depiction of transcription in the PURE system. (B) Imaging of transcription can be achieved using molecular beacons and aptamers. (C) Examples of the ways to monitor translation using fluorescent reporters. All elements are not to scale.

Mangos (Dolgosheina *et al.* 2014; Autour *et al.* 2018; Cawte *et al.* 2020), photostable Corn (Song *et al.* 2017), and multicolor Pepper (Chen *et al.* 2019; Tang *et al.* 2024) that, depending on the bound dye can emit cyan-to-red photons. Importantly, Pepper aptamer was imaged not only with conventional wide-field microscopy but also using structured illumination microscopy and two-photon confocal microscopy (Chen *et al.* 2019). The range of emission wavelength spectra of aptamers enables the imaging of multiple species of RNA being synthesized in parallel, for example, using a multimode microplate reader (Yan *et al.* 2024). Or aptamers can be designed in a way that they can bind the ligand and fluoresce only when another activator molecule is present, allowing for even more metabolic studies (Wang and Simmel 2023). The downsides of aptamer transcription imaging are related to their degradation by nucleases, limited dye diversity, low thermal stability, and background noise (Odeh *et al.* 2019). Nevertheless, aptamers have been extensively used to quantify transcription yields in cell-free expression systems. In particular, for PURE studies, a single Spinach aptamer sequence separated by a sufficiently long linker of 36 bases not to impede ribosomal activity or aptamer folding (Van Nies *et al.* 2015, p. 201) has been shown to be an effective transcription reporter (Van Nies *et al.* 2015; Doerr *et al.* 2019). Aptamers were also used to produce numerical models of transcription dynamics *in vitro* (Doerr *et al.* 2019; Zhao and Wang 2024). Moreover, applications of multiple repeats or tandem aptamers showcase a promising strategy to enhance the sensitivity and brightness of aptamers (Chinnappan *et al.* 2013; Zhang *et al.* 2015) for future synthetic cell studies.

## Imaging translation *in vitro*

The most commonly used approach for *in vitro* translation imaging is direct reporting with fluorescent proteins. This can be achieved using green fluorescent protein (GFP) (Prendergast and Mann 1978; Prasher *et al.* 1992; Chalfie *et al.* 1994) and its variants, such as enhanced GFP (eGFP) with superior fluorescent properties (McRae *et al.* 2005), superfolder GFP with better folding and higher stability (Pédelacq *et al.* 2006), and split GFP, split into two nonfluorescent fragments, thus making it apt for colocalization studies (Cabantous *et al.* 2005; Cabantous and Waldo 2006), or with other colorful fluorescent proteins (Rodriguez *et al.* 2017). The advances in this field over the past 30 years have made it possible to monitor the expression of several proteins and to track simultaneous transcription–translation *in vivo* and *in vitro*. Examples of the *in vitro* applications include using Spinach aptamer in combination with yellow fluorescent protein for quantitative description of the PURE system in GUVs (Van Nies *et al.* 2015; Doerr *et al.* 2019), transcription–translation coupling studies in lysate with AlexaFluor 488 molecular beacon and eGFP (Hansen *et al.* 2016b) and characterization of riboswitch functions in cell extract employing Mango-(IV) aptamer together with shifted GFP (Bains *et al.* 2023). Going forward with transcription–translation studies, careful consideration has to be given to the spectral separation of the two probes, DNA sequence design that does not impede ribosomal activity (Lentini *et al.* 2013), and fluorescent protein folding times, which cause maturation delay in translation reporting.

In some cases, expressed proteins can have a substantial effect on *in vitro* system metabolism and the properties of the GUV. Thus, the transcription could be tracked indirectly, such as in the case of the replication machinery production inside of the GUVs (Van Nies *et al.* 2018) and lipid synthesis (Scott *et al.* 2016) imaged using scanning confocal microscopy. Alternatively, reporters could

be incorporated into the produced protein, as in the case of FluoroTect™ GreenLys (Promega), which uses lysine-charged tRNA labeled with BODIPY®-FL to incorporate fluorescent amino acid into the protein sequence. There is limited literature on its use outside of imaging *in vivo*-produced proteins on a gel. Although the use of GreenLys can affect the functionality of protein when incorporated into functional domains and produce high background noise from the unincorporated amino acids, a poly-lysine appendix to the protein sequence in live bulk measurements could be a promising direction of research.

Finally, there is an option to deploy fluorescent labels that are associated with the expressed protein. SunTag is an approach that requires multiple peptide epitopes to be added to the protein sequence. Once the protein is expressed, these epitopes bind tags, comprised of single-chain variable fragment antibody, GCN4 peptide, and fluorescent protein, typically GFP. The multiplexing of 10–24 GFP copies on the SunTag scaffold enables long-term single-molecule imaging at the cost of substantial molecular weight (Tanenbaum *et al.* 2014). It can also carry a functional protein, for example, a DNA-interacting protein, instead of a fluorescent one and perform regulatory functions (Shakirova *et al.* 2020). This customizability makes SunTag another viable candidate for imaging transcription *in vitro* and reconstituting complex genetic networks in synthetic cells.

## Discussion and outlook

Having discussed the means of reconstituting and observing transcription–translation in GUVs, we now assess the main challenges ahead. The first one is the lack of standards when it comes to developing cell-free expression systems, which are developed with the maximization of the yield in mind. After all, the motivation behind the currently available commercial cell-free expression kits, such as PURE variants, was to produce high quantities of protein of interest from a single gene plasmid *in vitro* (Shimizu *et al.* 2001). This is a straightforward metric that is easy to conceptualize and optimize. However, the bottom-up synthetic cell would require a different type of transcription–translation system, optimized instead for a chromosome containing a complex gene network with embedded regulatory interactions necessary for ensuring metabolic activity and reconstituting the cell cycle (Olivi *et al.* 2021). It is a whole separate problem, identifying how such a cell-free expression system would differ in design and content, yet there have already been promising steps toward this, like in the case of lysate-based and commercially available myTxTl kit, adaptable to program gene circuits with sigma factors (Shin and Noireaux 2012; Garamella *et al.* 2016; Garenne *et al.* 2021b, 2019).

On top of that, the bottom-up approach entails the convergence of various building blocks, which are often at odds with each other. As more effort is put into synthetic biology, there will be both the demand for and the spark of innovation in biophysical techniques. One such innovative field could come out of disentangling transcription–translation decoupling, which is currently focused on combining the T7 transcription and *E. coli* translation machinery in a bottom-up PURE system. Some optimization has already been done to address the underlying issues of inefficient ribosomal usage (Doerr *et al.* 2019, 2021) by increasing ribosome recycling efficiency, reducing ribosome stalling, and raising the fraction of functional full-length protein through the addition of associated factors (Li *et al.* 2014, 2017). These attempts highlight the rational way forward. Cell lysate is a mixture of components just like the PURE system, yet the latter is 1–2 orders of magnitude less efficient,

even though the transcription in it is more rapid (Gregorio *et al.* 2019). This means that through the rational optimization of the solution composition, it should be possible to enhance the bottom-up system by an order of magnitude.

In practice, this would entail a crude, labor- and time-intensive exploration of a multiparameter space to optimize reaction performance. Among the parameters to explore are the components already included in the PURE system (Shimizu *et al.* 2005) and the additional elements taken from the list of lysate components (Foshag *et al.* 2018), but also the buffer condition components, such as crowding agents, that affect transcription–translation by changing the environment where the reaction is taking place (Deng *et al.* 2018). The number of parameters is limited but sizeable, and some of them are interconnected, such as the nucleotide and $MgCl_2$ concentrations (Kartje *et al.* 2021), while others could be assigned a fixed value, like using a single DNA per synthetic cell (Olivi *et al.* 2021). Together, the resulting list of parameters would include several hundreds of salts, proteins, and nutrients. The exploration of this vast set of conditions using manual labor could be a waste of funding and time for highly qualified scientific staff. Instead, we envision the use of automated pipetting robots, similar to how automatization is now employed in other industries involving precise manipulation, be it pharmaceutical or machinery manufacturing. Furthermore, the navigation through this parameter space would require an algorithmic approach, where the conditions of all the past experiments and their results are considered. Bayesian optimization, commonly used for global optimization without assuming any functioning form, has found an array of applications in the time of machine learning advances (Garnett 2023). This approach's focus on both random exploration of the uncharted areas of the parameter space and the narrowing down on the prospective regions to find the optimal conditions will make it useful for this task (Wakabayashi *et al.* 2022). Even then, the convergence of components, the scale of the challenge, and the potential benefits warrant this to be a very costly, large-scale, long-term project similar to the previous mega-projects of the past years.

For example, the Human Genome Project ('Human Genome Project Fact Sheet' n.d.) took from 1990 to 2003, with some work not being finished till 2022. As a result, genomes were sequenced for species of increasing complexity, starting with *Haemophilus influenzae* bacteria and culminating in the entire human genome. The project, with a total cost exceeding $3 billion, was a collaboration between 20 universities, laboratories, and companies from six countries. While the financial costs may seem immense, they were covered by the economic effects of the project, which led to the explosive growth of the bioinformatics field (Venter *et al.* 1998), produced unprecedented insights into our genetics used in therapeutics nowadays, and sparked the development of technologies.

Similarly, a synthetic cell project would be very costly, yet it would bring innovation to the fields of therapeutics, agriculture, and biomaterials (Khalil and Collins 2010). Moving toward completion would require international collaboration and long-term funding. For that reason, it is promising to see the increasing focus on synthetic biology in the last decades. The examples include institutes like JCVI (USA), collaborative networks like Build-A-Cell (USA), fabriCELL (UK), and SynCellEU (EU), and funded projects like BaSyC & EVOLF (NL) and MaxSynBio (DE). With the growing scale of these networks and projects, we could see significant progress toward understanding life as the convergence of building blocks, the development of new biophysical technologies, and the rapid growth of synthetic biology applications.

**Open peer review.** To view the open peer review materials for this article, please visit http://doi.org/10.1017/qrd.2024.27.

**Supplementary material.** The supplementary material for this article can be found at http://doi.org/10.1017/qrd.2024.27.

**Acknowledgements.** We express our gratitude to David Dulin for his extensive contribution to discussions on transcription–translation studies; to Kristina Ganzinger and Lori van Cauter for their collaborative spirit and knowledge-sharing on GUV production methods; to Christophe Danelone, Marileen Dogterom, Gijsje Koenderink, their lab members and other people involved in BaSyC for the workshops on the PURE system and insightful scientific meetings.

**Authors contribution.** VGB: writing and visualization, GJLW: writing and supervision.

**Financial support.** NWO funded the work as a part of Building a Synthetic Cell (BaSyC) project.

**Competing interest.** GJLW is a co-founder of LUMICK B.V., producing setups for single-molecule fluorescence studies of DNA–protein interactions.

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
