## [Reviewer Report]

In this review, the authors briefly survey the current state of the synthetic cell field. They start by describing currently used cell-free expression systems, followed by GUV production/encapsulation methods, and finally means to monitor processes in such systems. The article gives a good overview for newcomers in the field, but it is not entirely clear what the main scope of the article is – it seems that the authors wish to stress that advanced microscopy/single molecule methods could help in the engineering of synthetic cells?

Apart from this a few aspects seem to be missing and there are a number of language issues, specifically:

- the abstract somewhat tries to explain what the scope of the article is, but the introduction does not really do it – the final sentence “we put an additional emphasis on

techniques that allow single-molecule transcription tracking at low DNA copy numbers in GUVs” thus comes a little surprising (why “additional”, also there has been no discussion of any "emphasis” before)

- also it is arguable whether the article really describes “in detail” how PURE and cell lysate are used (here one would expect a list of components, etc.)

- the introductory paragraphs to each section should be shortened or completely erased. They are not necessary.

- why do you provide a table for the encapsulation method, but not, e.g., for the different expression systems?

- erase “In the next part of this review, we will discuss imaging probes for the detection of

transcription and translation in vitro.”

- while protein expression/encapsulation are standard aspects discussed in other articles, the “imaging” section seems to be the main difference to other such articles? Maybe this could be highlighted more?

- what seems to be missing in “imaging” are technical requirements such as types of microscopes (confocal?), immobilization of samples (?), concentrations of species (?), etc.

- The sentence “Aptamers (Ellington & Szostak 1990) are another type of on-when-bound probe” is not generally correct as most aptamers were developed as binders and not as sensors/probes. The authors focus on “fluorescence light up aptamers” (FLAPs), but there are other types.

- there are actually several recent publications on fluorescent aptamers that detect RNA, e.g., DOI: 10.1021/acssynbio.3c00426, 10.1016/j.chempr.2024.03.015, 10.1002/ange.202302858

- arguably, among the main challenges for the field are a metabolism, self-regeneration/production of ribosomes, self-replication, which are not mentioned at all

- p.1: “that remains” → “that remain”

- p.1 : “extremeties” → “extremes”

- p.1: “rationalistic” → “rational”

- p.3: “RNase and protease” → “RNases and proteases”

- p.6. “lays somewhere” → “lies somewhere”

---

## [Reviewer Report]

This perspective offers an examination of the challenges in transcription-translation systems within bottom-up synthetic biology, focusing on synthetic cell construction. The authors outline fundamental issues in developing synthetic cellular systems, specifically highlighting the PURE system and cell lysate-based approaches. The discussion on encapsulation methods, particularly within GUVs, addresses key challenges and potential advancements. However, after reviewing the manuscript, I found it lacking in strong points or useful insights across the listed aspects. The overall structure is somewhat loose, lacking a strong narrative and clear logic, which are essential for a perspective paper.

Detailed Comments:

1. Transcription-Translation Section: As mentioned, there is a lack of clear logic and a loosely organized structure in this section. A more detailed comparison between the PURE system and cell lysate systems could improve the reader’s understanding of why certain applications may favor one approach over the other. Including specific use cases where one method clearly outperforms the other could also provide practical insights.

2. GUV Encapsulation Efficiency: While the comparison of various GUV production methods is informative, including additional quantitative data, where available, on encapsulation efficiency and yield consistency would add value. Such data would enable readers to better evaluate the scalability and feasibility of these methods in practical applications.

3. Outlook on Future Innovations: The authors briefly mention potential future advancements, such as the need for improved standards and multi-component optimization. Expanding this section to include specific anticipated technological or methodological breakthroughs could enrich the outlook and inspire further research.

4. Addressing Limitations in Real-Time Imaging: Although the paper discusses various imaging techniques, a deeper exploration of the limitations, such as in molecular beacon and aptamer applications, would provide a more balanced view. Potential solutions or alternative approaches to overcome these limitations could enhance the practical application of these imaging techniques in synthetic cell development.

Minor Points:

• Page 3, First Paragraph: The phrase “commonly E. coli lysate is relatively easy to prepare” is somewhat misleading. Although detailed protocols and commercially available kits exist, establishing a tailored cell-free expression system remains a complex task for a standard biochemistry lab. Numerous publications provide valuable insights into this process, which were not included in this manuscript.

• Page 11: Several variants of green fluorescent protein are mentioned. The authors should exercise caution and accuracy in citing them, including specific forms such as eGFP and shifted GFP.